# XGBLoc: XGBoost-Based Indoor Localization in Multi-Building Multi-Floor Environments

**DOI:** 10.3390/s22176629

**Published:** 2022-09-02

**Authors:** Navneet Singh, Sangho Choe, Rajiv Punmiya, Navneesh Kaur

**Affiliations:** 1Department of Information, Communications, and Electronics Engineering, The Catholic University of Korea, Bucheon-si 14662, Korea or; 2Center for Distance and Virtual Learning, University of Hyderabad, Hyderabad 500046, India or

**Keywords:** RSSI fingerprints, WiFi, indoor localization, XGBoost, classification, regression, hyper-parameter tuning, labeling

## Abstract

Location-based indoor applications with high quality of services require a reliable, accurate, and low-cost position prediction for target device(s). The widespread availability of WiFi received signal strength indicator (RSSI) makes it a suitable candidate for indoor localization. However, traditional WiFi RSSI fingerprinting schemes perform poorly due to dynamic indoor mobile channel conditions including multipath fading, non-line-of-sight path loss, and so forth. Recently, machine learning (ML) or deep learning (DL)-based fingerprinting schemes are often used as an alternative, overcoming such issues. This paper presents an extreme gradient boosting-based ML indoor localization scheme, simply termed as XGBLoc, that accurately classifies (or detects) the positions of mobile devices in multi-floor multi-building indoor environments. XGBLoc not only effectively reduces the RSSI dataset dimensionality but trains itself using structured synthetic labels (also termed as relational labels), rather than conventional independent labels, that classify such complex and hierarchical indoor environments well. We numerically evaluate the proposed scheme on the publicly available datasets and prove its superiority over existing ML or DL-based schemes in terms of classification and regression performance.

## 1. Introduction

Indoor localization (IL) is tremendously popular in recent years due to the growing availability of the Internet of Things (IoT). As a result, there has been an accelerating demand for highly accurate and low-cost IL [1]. However, GPS signals are almost unreliable in indoor settings, and therefore, researchers have explored other techniques using wireless signals, visible light signals, magnetic field signals, and inertial sensors signals [2,3,4,5,6,7,8]. A wireless positioning system (WPS) uses indoor wireless signals to track down the location of the target objects [4]. The WPS techniques are classified as either geometric-based or fingerprinting-based approaches. The geometric approaches use the estimated angles and distances between the wireless access points (WAPs) and the target devices (or mobile devices or subscribers) for IL classification. However, those approaches still suffer from outlier distortion such as non-line-of-sight and indoor multipath faded signals. Furthermore, the resulting significant communication overhead and the need of good synchronization circuits between devices (transmitters and receivers) increase the overall cost of geometric-based WPS [9,10].

Hence, the fingerprinting-based approaches are often used [11] as an alternative. A fingerprint (fi) is an array of wireless-received signal strength indicators (RSSI) from in-range WAPs that indicates the measured radio signal magnitude at a specific reference point (RP). The fundamental use of fi is to compare (and match) the real-time received signal patterns with the pre-collected signal patterns [11]. The globally deep penetration of WiFi infrastructure makes WiFi RSSI signals one of the most popular (and easy) WPS fingerprints, especially after the introduction of ML/DL techniques.

Broadly, there are two phases in the fingerprinting-based localization using ML/DL: offline (or training/testing) phase and online (or real implementation) phase. First, in the offline phase, fi (WiFi RSSI values) are obtained from *n* predefined RPs such that a radio map (database) is constructed. This radio map is used to train and test designed ML/DL models. During the training/testing phase, the ML/DL models can be viewed as one of the global approximation algorithms that maps the RSSI observations to the locations of devices with the help of pattern recognition techniques. Second, in the online phase, this trained model is used to predict the locations of the devices based on real-time collected RSSI data [12].

In the literature, we can broadly categorize IL environments into the following three different types, i.e., single building single floor, single building multi-floor, and multi-building multi-floor, for which fingerprinting-based localization schemes using ML/DL have been applied. However, in our previous work [12], existing localization schemes are mostly focused on single building single floor or single building multi-floor indoor environments using custom or private datasets. Hence, those existing schemes have a limited performance for the classification of multi-building multi-floor indoor environments [13]. One of the main reasons is that existing schemes just use independent or non-relational labeling (NRL) for the representation of such complex and hierarchical indoor environments, where relationship between a building and its corresponding floors is neglected—please see Section 2 for further explanation regarding the shortcomings of existing schemes. Therefore, in the paper, we propose an ML scheme using relational labeling (RL), where unique synthetic labels (see Section 3.2 for the detail explanation) representing the relationship between building IDs and floor IDs are defined and used, such that its classification performance is improved.

From several data-science competitions, it is well-known that extreme gradient boosting algorithm (XGBoost) is superior to existing ML/DL algorithms [14] in terms of speed and performance. However, during the related literature review, we have noticed a lack of gradient-boosting-based localization algorithms. Hence, in the paper, we propose an IL algorithm using XGBoost [14,15] (which is simply termed as XGBLoc). In that algorithm, a principal component analysis (PCA)-based preprocessing module [16,17] is introduced that reduces the high number of dimensions of the input RSSI signals, decreases the impact of outliers, and resolves the overfitting issues. As a result, XGBLoc relatively has a better localization performance compared to existing ML/DL schemes (see Section 2), while keeping a lower design complexity.

The main contributions of this work are summarized as follows:Create a novel and uniquely-combined synthetic label (also called relational label) directly associating a building ID with a floor ID in a multi-building multi-floor environment. Using this relational labeling (RL) rather than existing non-relational labeling (NRL; or independent labeling, where building ID and floor ID are separately dealt with), the presented ML-based classification model predicts target locations in such complex hierarchical environments accurately and consistently.Propose an XGBoost-based IL method using RL, termed as XGBLoc, that improves localization accuracy in the multi-building multi-floor environments. XGBLoc is especially good for the classification and regression over such hierarchical and complex indoor environments. XGBLoc employs PCA both for dimensionality reduction and input dataset (i.e, RSSI fingerprints) denoising. After PCA transformation, XGBLoc is trained and tested to predict target location with an improved accuracy over dynamic (mobile) indoor channel conditions including multipath fading.Evaluate XGBLoc over the following three publicly available datasets: UJIIndoorLoc [18], Tampere [19], and Alcala [20]. Simulation results validate that the proposed scheme has a superior localization performance over existing ML/DL schemes (see Section 4 for the details) especially at the perspective of localization accuracy as well as system complexity.

The article is organized as follows. Section 2 surveys recently-published related articles and discusses the shortcomings of those existing ML or DL schemes. Section 3 presents details about the methodology of proposed system model XGBLoc, including proposed system architecture, input and output specification, objective function, and RSSI data and preprocessing. In Section 4, we discuss the performance analysis of the proposed model. Finally, we present the conclusion in Section 5.

## 2. Related Works

Recently, fingerprint-based ML localization approaches such as *K*-nearest neighbor (KNN) [21], WKNN [22], and support vector machine (SVM) [23] are introduced. However, most of them are not well suited to the dynamic nature of multi-building multi-floor environments such that their positioning performance falls short of actual requirements [24]. Therefore, many researchers have also tried DL-based approaches as an alternative. In [25], the authors proposed a recurrent neural network (RNN)-based positioning system for localization of the devices, where the multi-output Gaussian process is applied to establish the correlation among RSSI values from closely-deployed multiple access points for better performance. The experiments show that their RNN system can achieve 100% and 94.20% accuracy in building and floor classification, respectively. Another RNN-based indoor localization scheme for multi-building multi-floor environments is proposed in [26]. This scheme produces predictions from building to floor to location in a sequential manner. The authors in [26] reported that their proposed scheme can achieve 100% and 95.24% accuracy for building and floor classification, respectively. CNNLoc [13] is a multi-building and multi-floor IL system that uses WiFi fingerprints. It uses a stacked autoencoder (SAE) to extract specific features from raw RSSI fingerprints and employs a CNN to achieve high accuracy during the online phase. As a result, CNNLoc can achieve accuracies of 100% and 96%, respectively, for building and floor classification. In [27], the authors proposed DeepLocBox (DLB) which predicts the position of the users using a single DNN model. The experiments show that DLB can achieve 99.64% and 92.62% of accuracy for building and floor classification, respectively, in multi-building multi-floor environment. In a CNN-based system [28], the authors constructed a 2-D virtual radio map from 1-D WiFi RSSI values and then implemented a CNN system to handle 2-D radio map inputs. Therefore, this system can learn the topology of an RSSI-based radio map such that it achieves 95.41% accuracy for predicting the building and floor numbers [18]. In [29], the authors proposed a fingerprinting-based multi-cell encoding learning (m-CEL) technique for position estimation within large-scale indoor environments. This m-CEL technique deals with building and floor classification as a solution of multi-task learning problems within a single forward pass network. Their proposed 2-D CNN model using m-CEL can achieve 95.3% of accuracy in building and floor classification.

From several data-science competitions, it is well-known that gradient-boosting techniques are superior to existing ML/DL algorithms [14,30] in terms of speed and performance. However, through the literature review, we have noticed a lack of gradient-boosting-based localization algorithms. Hence, in the paper, we propose an IL algorithm using XGBoost [14,15], which is simply termed as XGBLoc. A summary of recent related works is shown in Table 1.

### Shortcomings of Existing Schemes

In the literature review, we have found that most of existing ML/DL schemes have the following two shortcomings [13,25,26,27,28,29].

First, those existing ML/DL schemes (to our best knowledge) use NRL in multi-building multi-floor IL environments, where targeting objects, i.e., buildings and floors, are predicted (or classified) independently. That is, they do not take direct relationships between buildings and their floors into account during the ML/DL training & testing process. For example, in Figure 1 [left], building 1 has floor IDs 1, 2, and 3 and building 2 has floor IDs 1, 2, 3, and 4. During the floor classification process, when the existing model deals with floor ID 2, it creates confusion regarding from which building this floor ID belongs to since floor ID 2 is present in both buildings. As a result, those existing schemes using NRL do not easily distinguish between different floors (with the same floor label) of different buildings such that classification performance is degraded, especially in multi-building multi-floor environments. Furthermore, those existing schemes using NRL request two distinct models for the ML/DL-based localization process, i.e., one for building classification and the other for floor classification, such that system complexity may increase almost twofold.

Second, from the literature review, it is evident that most of existing ML/DL-based methods still have the overfitting or underfitting problems [31] and suffer from selection bias. Hence, those existing schemes request extensive hyper-parameter tuning for the training over multi-building multi-floor environments in order to mitigate such overfitting & underfitting, and selection bias issues. As a result, this extensive tuning process may increase the system complexity of existing schemes further.

Therefore, in the paper, we introduce relational labeling (RL) rather than NRL for the improved classification over such hierarchical IL environments and present a novel XGBoost-based ML method using RL, simply termed as XGBLoc, such that all above-mentioned shortcomings are well dealt with. The details of the proposed system model XGBLoc are given in the following sections.

## 3. Proposed Methodology

In this section, we first present the system architecture of the proposed IL model XGBLoc, and then introduce the input & output specification, objective function of the proposed model, and finally address RSSI data and preprocessing.

### 3.1. Proposed System Architecture

The system architecture of the proposed IL model, termed as XGBLoc, is shown in Figure 2. It consists of two phases: an offline phase and an online phase. In both phases, we preprocess the input data by removing missing data, normalizing the RSSI values between 0 and 1, and applying PCA transformation to these normalized values for the dimensionality reduction. In the proposed model XGBLoc, XGBoost is used both for classification predicting target location labels and for regression estimating the 2-D coordinate target positions. First, in the offline phase, the presented model is trained using the preprocessed data. For the classification process, we create synthetic relational labels by concatenating building ID with their respective floor IDs and train XGBLoc using those labels. Moreover, for the 2-D regression process, we use latitude and longitude values of RPs. Consequently, during the online phase (as seen in Figure 2), the trained model first receives the RSSI values, and then preprocesses those values accordingly, and finally predicts the required localization output using the preprocessed data.

### 3.2. Input and Output Specification

In order to devise our model architecture, we first need to acquire a general IL dataset. In any given indoor localization dataset, fingerprints are usually defined as follows: fi={R,long,lat,fl,bi}. Depending on the dataset, the input and output can vary (such as absence of multiple buildings, etc). Therefore, we try to generalize the proposed model as much as possible.

Throughout the paper, we mainly consider the UJIIndoorloc dataset [18], whose fingerprint elements are listed in Table 2. In the training phase (i.e., offline phase), the RSSI value *R* is combined with labels (long,lat,fl,bi) to form a sample (R,long,lat,fl,bi), where (long,lat) is coordinates of landmarks (RPs) that are situated in building ID bi on floor ID fl. In the trained phase (i.e., online phase), the estimated output of the presented model, i.e., either ($\hat{fl}$, $\hat{bi}$) for classification tasks or ($\hat{long}$, $\hat{lat}$) for 2-D regression tasks, is obtained. For that estimation, real-time measured RSSI data *R* are used.

Currently, during data collection in the offline phase in a hierarchical smart environment, raw WiFi RSSI measurements are taken from a particular landmark or a RP, which is present in floor *a* of building *b*. In the offline phase, most of the conventional classification schemes use NRL, as shown in Figure 1 [left], where floor ID fl=a and building ID bi=b are treated as distinct independent variables [18]. In contrast, the proposed scheme employs RL, as shown in Figure 1 [right], where a relational label relating those two IDs, i.e., fl & bi, is defined and used for the localization over the multi-building multi-floor environment. In the RL, we exploit one-to-many relationship cardinality [32] that maps each of every building with its corresponding floors, where each relationship is represented by a uniquely assigned relational label (or relational identification number), as shown in Figure 1 [right].

For example, if building 1 has four floors, we can obtain a uniquely-assigned 4-digit relational label for each floor by applying one to many mapping, such as 1001, 1002, 1003, and 1004, where the first digit represents building numbers and last three digits represent floor numbers. By doing so, we establish a dependency of floor IDs on building IDs, which enables to classify building and floor together using only a single classification model. Then this classification model easily extracts floor IDs and their corresponding building IDs from those relational labels. Resultant output eliminates the chances of obtaining wrong combinations between floor IDs and building IDs. Thereby the proposed model reduces the false mismatch of floor IDs and building IDs during the prediction, as shown in Figure 1 [right]. This implies that for a classification problem the resultant output of the proposed scheme (using that single classification model) correctly estimates the relationship of a building number and its corresponding floor numbers. Furthermore, for a 2-D regression problem, the output of the proposed scheme ($\hat{long},\hat{lat}$) represents the user’s estimated location in the multi-building multi-floor IL environments.

Conventional ML/DL schemes that mostly use NRL labels classify building IDs well since there is a distance of at least a few meters away between buildings, which means that WiFi signals of a building suffer less interference from the other nearby buildings. In addition, in a multi-building multi-floor environment, each building has a unique building ID; for example, if there are four buildings, the building IDs could be given as 1, 2, 3, and 4, respectively. As a result, those existing classifiers easily learn using NRL labels and associate a WAP with its corresponding building, resulting in improved performance at the perspective of building classification. However, when it comes to floor classification, it is a different story. The WiFi signal interference between adjacent floors in the same building is relatively high, and the floor ID might not even be unique across the entire multi-building complex. Hence, those existing classifiers may result in a degraded floor classification performance.

Differing from NRL, RL allows the classifier model to easily associate a building with its floors because each floor has its unique floor ID. Moreover, XGBoost that easily deals with such tabular data learning is a good candidate for classifying RL labels. Therefore, the proposed ML algorithm XGBLoc distinguishes between the floors of different buildings well over such complex hierarchical environments, resulting in increased classification performance.

### 3.3. Objective Function of Proposed XGBoost-Based ML Model

In the proposed model, XGBoost leveraging a gradient boosting framework is used as the foundation of our IL algorithm. It is well-known that XGBoost, a decision tree-based ML algorithm, is good for the learning of structured data. Some of the key reasons we choose XGBoost over other ML and Dl algorithms such as random forest (RF), DT, GBDT, and CNN are summarized as follows [30,33]:With a similarity score, XGBoost prunes the tree. It calculates the node’s gain as the difference between the node’s similarity and the children’s similarity score. When the node’s gain is found to be nominal, it simply stops constructing the tree to a greater extent.In real-world applications, classification performance, computational cost, and hyperparameter optimization are critical factors of choosing a good classifier. In the paper, we choose XGBoost as such a good candidate for the IL applications where a target object is localized over a multi-building multi-floor environment. Moreover, when compared to traditional ML/DL classifiers, XGBoost is capable of handling real-time data with many variations.Especially, we use relational labels representing that hierarchical environment in a given dataset and train XGBoost using such translated tabular data. The proposed algorithm, simply termed as XGBLoc, performs better on those tabular data even with fewer data samples, when compared to other ML/DL algorithms.Furthermore, to deal with over- and under-fitting, those existing ML/DL algorithms require extensive hyperparameter tuning. For instance, ML algorithms such as RF, KNN, and so forth require longer computational time. Furthermore, DL algorithms require a large number of data samples to perform well. However, XGBLoc requests a relatively simple hyperparameter tuning.

XGBoost, which is an ensemble tree approach, employs the gradient descent architecture boosting weak learners. Compared to typical gradient boosting schemes, XGBoost enhances the underlying gradient boosting framework further with system optimization and algorithmic improvements, which includes hardware optimization, parallel tree building, efficient handling of missing data, tree pruning using the depth-first approach, and regularization through both LASSO (L1) and Ridge (L2) for avoiding overfitting. A regularized (L1 and L2) objective function that has a convex loss term and a model complexity penalty term is minimized by XGBoost [14]. The training proceeds iteratively until the final prediction results are obtained, while new trees that may reduce the residual errors of previous trees are continually inserted. Table 3 shows the symbols used for defining the objective function of proposed ML model.

Consider a fingerprint dataset that consists of *N* numbers of samples $D=\left(R_{1},l_{1}\right),\left(R_{2},l_{2}\right),$$\left(R_{3},l_{3}\right),\cdots ,\left(R_{N},l_{N}\right)$, where $R_{i}=\left[r_{1}^{i},r_{2}^{i},r_{3}^{i},\cdots ,r_{M}^{i}\right]$ is the vector of received *M*-dimensional RSSI values of the *i*th RP and $l_{i}=\left[x_{i},y_{i}\right]$ is the position of the *i*th RP. The dataset *D* can be decomposed into two subsets $SD_{1}=\left(R_{i},x_{i}\right)$ and $SD_{2}=\left(R_{i},y_{i}\right)$ for the modeling. For $SD_{1}$ and $SD_{2}$, XGBoost is used to predict $\hat{x_{i}}$ and $\hat{y_{i}}$, respectively. Putting them together can be termed as an estimated position coordinate $\hat{l_{i}}=\left[\hat{x_{i}},\hat{y_{i}}\right]$. Considering the subset $SD_{1}$ as an example, where *K* trees are assumed to have been trained, the predicted output for the *i*th sample is
(1)$${\hat{x}}_{i}=\sum_{k=1}^{K}f_{k}\left(R_{i}\right),$$
where $f_{k}\left(R_{i}\right)$ is the predicted output of the *k*th tree for the sample $R_{i}$. The objective function can be modeled as
(2)$$Obj=\sum_{i=1}^{N}L\left(x_{i},{\hat{x}}_{i}\right)+\sum_{k=1}^{K}\omega \left(f_{k}\right),$$
where $L(\cdot_{i},\cdot_{i})$ represents the *i*th loss function and $\omega (f_{k})$ is the complexity/regularization term of the *k*th tree. For more mathematical details, please refer to [14].

XGBoost supports hyperparameter tuning in order to tackle underfitting or overfitting problems. That is, the proposed scheme XGBLoc is tuned with hyperparameter tuning such that system performance is improved. Table 4 lists hyperparameters of XGBLoc and its corresponding default values. Note that the value of “loss function” needs to be set “multi:softprob” for a multi-class classification problem or “reg:squarederror” for a regression problem, respectively.

### 3.4. RSSI Data and Preprocessing

According to the log-normal propagation model, the WiFi RSSI values measured degrade exponentially as the distance between a transmitter and a receiver increases [34]. Moreover, in real-world scenarios, WiFi RSSI not only suffers from the interference by other radio signals but from the multipath caused by complex and dynamic indoor environments. All of these issues increase non-linearity and uncertainty of the WiFi RSSI signals. Another important aspect is the sparsity of raw WiFi RSSI data. Many WAPs including commercial APs and private Internet-connected smart IoT APs are installed in multi-floor buildings. In reality, at any particular reference point (RP), not all the APs can be scanned because an AP cannot cover the entire indoor environment. For instance, in the UJIIndoorLoc dataset [18], only about 190 APs among a total of 520 APs were scanned in a floor. That is, a user (or a device) cannot sense RSSI from some distant APs such that RSSI values of those APs are recorded as empty (’NA’), as shown in Figure 3.

Generally, these empty values are replaced with the minimum present RSSI value. For instance, in our evaluation of the UJIIndoorloc dataset, we replace those empty values with −98 dBm. However, sparsity remains a cause of concern because a large portion of that data is replaced with −98 dBm. Therefore, we need to reduce the dimensional sparsity. This dimensionality reduction will also help to reduce computational load and mitigate noise effects. Before reducing the dimensional sparsity of the data, we also need to bring RSSI values to a common scale because every vendor of WiFi may have different scales representing the RSSI values. To remove this heterogeneity from the data, we use a ZeroToOneNormalized technique [35], that gives the scaled RSSI values *x* as follows:(3)$$x=\left\{\begin{array}{cc} 0, & RSSI_{m}=100\\ \frac{RSSI_{m}-min}{-min}, & -98\leq RSSI_{m}\leq 0 \end{array}\right.$$
where $RSSI_{m}$ is the *m*th WAP’s RSSI value, min is the minimum value of RSSI in the offline fingerprint database, and the value of 100 is used to indicate that no AP was detected. Other than removing the effect of heterogeneity of devices, the primary purpose of normalization is to modify numeric column values in the dataset such that the standard scale is utilized without distorting or losing the information distribution. Each iteration of the XGBoost algorithm optimizes the samples according to the residual error such that the algorithm’s bias will decrease, forcing it to be less sensitive to the outliers and noise. Then, the presented scheme employs PCA [36] to extract some important features from sparse and raw WiFi RSSI values, while reducing the data dimensionality and decreasing impact of outliers.

PCA is a popular tool in current ML because it is a simple, non-parametric method for extracting meaningful information from complex datasets [36]. In addition, PCA provides a pathway for reducing a complex high-dimensional dataset to a lower dimensional one with minimal effort [37]. The fundamental advantage of PCA is that it quantifies the value of principal components in representing the variability of a dataset. The analysis of variance along with a small number of principal components (i.e., less than the number of measurement types) offers a meaningful representation of the entire dataset [36]. For instance, for the UJIIndoorLoc dataset [18], Figure 4 shows that the first 100 eigenvectors (i.e., principal components) could explain roughly 90% of the explained variance ratio (EVR). Algorithm  1 shows the pseudo code of the proposed XGBLoc scheme reflecting system methodology above mentioned in this section.
**Algorithm 1** Pseudo code of proposed XGBLoc scheme**Input:***Dataset $D=(R_{i},l_{i})$, fl, bi, long, and lat.***Output:***Predicted Location $\hat{l}$*1:      **while**
l← (bi, fl) **do**2:         RL←OneToMany(bi,fl)3:         *Set RL as labels.*4:      **end while**5:      **if**
l←(long,lat)
**then**6:         Skip Step 1 to 4.7:         *Set (long, lat) as labels.*8:      **end if**9:      **if**
(RSSI←100) **then**10:        $R_{i}.replace\left(RSSI,-98\right)$11:    **end if**12:    *$Normalized(R_{i})$←$ZeroToOneNormalized(R_{i})$.*13:    $R^{‘}$←PCA(Normalized(R))14:    *Set Dataset $D(R_{i},l_{i})$← Dataset $D^{‘}\left(R_{i}^{‘},l_{i}\right)$*15:    *training set, testing set, validation set ←Split(DatasetD)*16:    **if**
RL←labels
**then**17:       *Train classification XGBLoc model.*18:       *Tune Hyperparameters* **return** *Predicted location $\hat{l}$: Symbolic location.*19:       **return**
*Predicted location*
$\hat{l}$*: Symbolic location.*20:    **else** {(long,lat)←labels}21:       *Train regression XGBLoc model.*22:       *Tune Hyperparameters*23:        **return** *Predicted location $\hat{l}$: Physical location.*24:    **end if**

## 4. Performance Analysis

In this section, we numerically evaluate the proposed PCA-XGBoost scheme for indoor localization, termed as XGBLoc, and compare it with state-of-the-art approaches in terms of the classification and regression performance. Herein, we first and mainly use the UJIIndoorLoc dataset (multi-building multi-floor) for both classification and regression analysis over such hierarchical IL environments. In that localization analysis, XGBLoc that uses RL instead of NRL is compared to existing ML/DL schemes and is proved to be good for classification. For further evaluation of XGBLoc, we also use the Tampere (multi-floor) and Alcala (corridor) datasets. We implement the presented scheme using Python-3.8.5 on a laptop with AMD Ryzen 5900HS and RTX 3060 GPU.

### 4.1. Results on Benchmark UJIIndoor Dataset

In this subsection, we summarize the experimental results of XGBLoc on the UJIIndoorLoc dataset [18]. This dataset contains 21,048 fingerprint samples from three buildings, each one having different number of floors. For performing our experiments, we split the dataset into three subsets with the ratio of 80/15/5%: training, validation, and testing. As a result, the training, validation, and testing subsets contain 16946, 2991, and 1111 samples, respectively.

#### 4.1.1. Effect of Explained Variance Ratio on Performance

We have tested the proposed scheme XGBLoc with respect to different values of EVR and evaluated its optimal performance on the following two tasks, i.e., classification accuracy analysis and 2-D (long,lat) position error analysis, whose results are summarized in Table 5. From Figure 4, we can see that the top 100 principal components explain 90% of EVR. It indicates that through the PCA-based preprocessing, we may reduce the dimensionality of the input dataset up to 100 from 520. From Table 5, we can see that the classification performance converges to above 99% for EVR = 0.9 and the 2-D regression performance to its best value for EVR = 0.8.

#### 4.1.2. Effect of Hyperparameter Tuning on Performance

XGBoost supports hyperparameter tuning that helps in mitigating overfitting and underfitting problems. It implies that the performance of the proposed XGBoost-based system model, termed as XGBLoc, could be improved with an adequate hyperparameter tuning. Figure 5 shows the effect of different learning rates and n_estimators (which indicates the number of trees to be generated) on the classification accuracy of the proposed model. The overall trend shows that classification accuracy increases with the increase of n_estimators with different learning rates. From Figure 5, we can confirm that the classification accuracy of XGBLoc using the UJIIndoorLoc dataset is improved as n_estimators increase and converged to its maximum (99.2%), especially when n_estimators ≥ 500 and the learning rate is kept at 0.1. Table 6 also shows hyperparameter settings yielding an optimal performance of XGBLoc. It ensures that two different localization tasks, i.e., classification or regression, may need different hyperparameter tuning.

#### 4.1.3. Localization Performance Comparison

We have compared localization performance of the proposed scheme and the existing benchmark schemes over the same multi-building multi-floor IL environment, which include MOSAIC [38], 1-KNN [18], 13-KNN [39], DNN [40], 2D-CNN [28], scalable DNN [41], and CNNLoc [13]. Most of those benchmark schemes use non-relational labeling (NRL), where building IDs and floor IDs are independently labeled (see Figure 1 [left]). Hence, for the NRL classification over such complex IL environments, those existing schemes utilize the two separate classification models: one for building and the other one for floor. Those two classification models require the following separate processes per model: training, hyperparameter tuning, and testing. Hence, some additional complexity including more computational resources is not avoidable. Table 7 shows that existing schemes perform well for building classification with accuracy of 98% to 100% [13] but have some performance variation for floor classification. Specifically, whereas CNNLoc achieves the highest 96.03%, 1-KNN achieves the lowest 89.95% in terms of floor classification accuracy.

We have tested the proposed scheme XGBLoc, where a uniquely-combined synthetic label (also called a relational label) is assigned to each floor of different buildings (see Figure 1 [right]). Results in Table 7 show that XGBLoc achieves 99.2% of accuracy in terms of floor classification over the same multi-building multi-floor IL environment. Note that, unlike existing ML/DL-based schemes, XGBLoc requires only a single classification model that classifies buildings & floors together at the same time; thus its overall complexity could be reduced, even if the given dataset UJIINdoorLoc is obtained over a modest environment conditions.

We have also compared 2-D (long,lat) mean positioning error of XGBLoc with other WiFi fingerprint-based schemes such as KNN, WKNN, RF, CNNLoc, HybLoc [42], and CCpos [35], whose results are shown in Table 8. It verifies that XGBLoc with 4.93 m of positioning error outperforms existing schemes.

### 4.2. Results on Additional Datasets

To evaluate the scalability, adaptability, and robustness of XGBLoc further, we have tested its performance over the other two public datasets: Tampere [19] and Alcala [20]. Note that the Tampere dataset contains data from single building multi-floor environment and the Alcala dataset contains data from a corridor of the building.

#### 4.2.1. Results on Tampere Dataset

The Tampere dataset contains 4648 fingerprint samples collected from 992 WAPs in a Tampere university’s building with multiple floors. In the preprocessing, the default value of missed RSSI is set to be 100. Floor classification results of XGBLoc and various benchmark schemes including CNNLoc [13] are compared each other in Table 9. In that comparison, XGBLoc outperforms those benchmark schemes by 3% up to 15%. In such performance evaluation, we find out that value changes of a few hyper-parameters is enough for the proposed scheme. In the 2-D (long,lat) regression analysis, the proposed scheme is also dominant over those existing schemes (whose results are omitted in Table 9 for simplicity). Note that XGBLoc is able to achieve a 2-D mean positioning error of 7.02 m, which is much less than CNNLoc (whose result 10.88 m [13] is the best among existing schemes listed in Table 9).

#### 4.2.2. Results on Alcala Dataset

To further evaluate the regression performance of XGBLoc, we have also considered Alcala tutorial 2017 dataset [20]. This database could be used as an alternate of the UJIIndoorLoc dataset [18] for regression analysis. It is a relatively small dataset covering a corridor of the School of Engineering of the University of Alcala. It contains a total of 1075 fingerprint samples collected from 152 WAPs, whose values have a range from −99 dBm (extremely poor signal) to 0dBm (strongest signal), and the value 100 indicates undetectable WAP signals. Experimental results show that XGBLoc outperforms most of other WiFi fingerprint-based schemes such as KNN, WKNN, SVM, RF, CCpos and CNNLoc [13,35]; the results of 2-D mean positioning error of respective schemes are compared in Table 10. Although CCpos [35] could achieve higher accuracy than the proposed scheme, the difference is about 45 cm.

## 5. Conclusions

We have proposed a XGBoost-based ML model using WiFi fingerprinting, termed as XGBLoc, that localizes target device(s) over multi-building multi-floor environments. By employing the PCA-based preprocessing, XGBLoc extracts core important features from a WiFi fingerprint dataset while handling sparsity, reducing dimensionality, and removing noise. For the representation of multi-building multi-floor environments, we have defined relational labeling (RL) instead of existing non-relational labeling (NRL) and applied it to XGBLoc. As a result, while XGBLoc has a smaller system complexity, it gives a better classification and regression performance over existing ML/DL schemes. We have evaluated XGBLoc on three different publicly available datasets: UJIIndoor, Tampere, and Alcala. The performance results show that XGBLoc can achieve classification accuracy of 99.2% and 97.03% on the UJIIndoorLoc and Tampere datasets, respectively. Moreover, XGBLoc can locate target devices with an average positioning error of 4.93 m, 7.02 m, and 1.5 m on UJIIndoor, Tampere, and Alcala, respectively. Those results also show that the proposed XGBoost-based scheme provides a higher degree of scalability and robustness, and keeps a better balanced trade-off between model complexity and classification performance, when compared to existing ML/DL schemes.

In the future, we will expand our study further by collecting measured data from relatively large-scale multi-building multi-floor environments and then testing the behavior of the proposed scheme on it. Additionally, the proposed scheme will be expanded for achieving low-latency user localization, while preserving the user privacy and security, which could help develop the location-based access control and resource management system for smart factories.

## Figures and Tables

**Figure 1 sensors-22-06629-f001:**
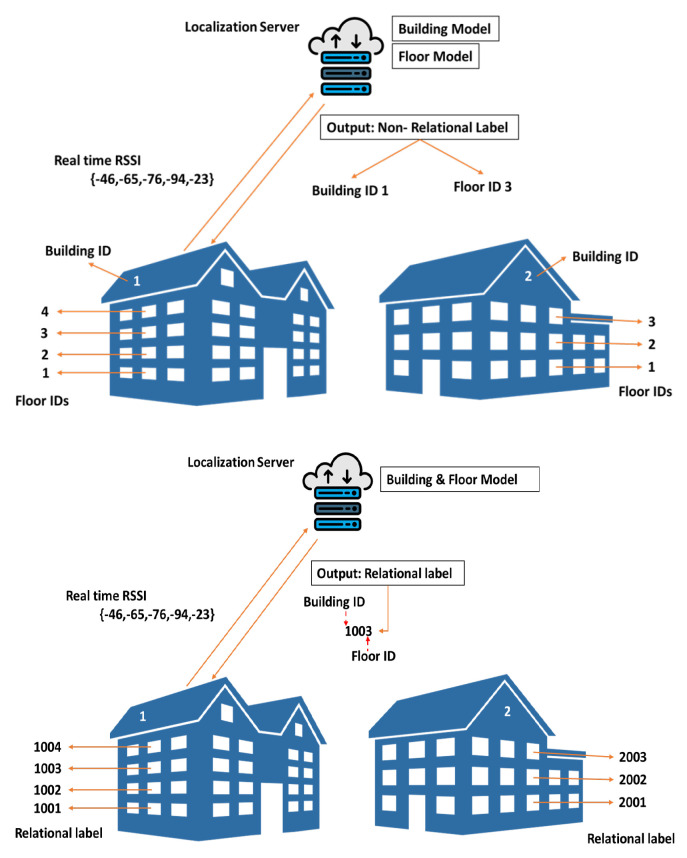
Conventional non-relational labeling (NRL) [left] and proposed relational labeling (RL) [right] for a multi-building multi-floor environment example.

**Figure 2 sensors-22-06629-f002:**
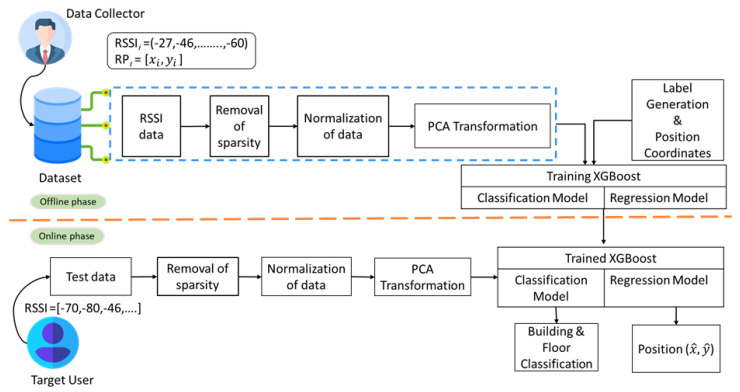
Architecture of proposed system model XGBLoc.

**Figure 3 sensors-22-06629-f003:**
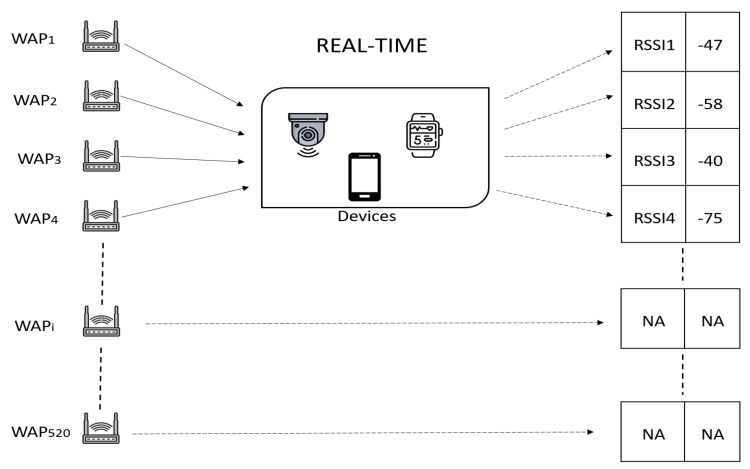
Sparsity in RSSI values.

**Figure 4 sensors-22-06629-f004:**
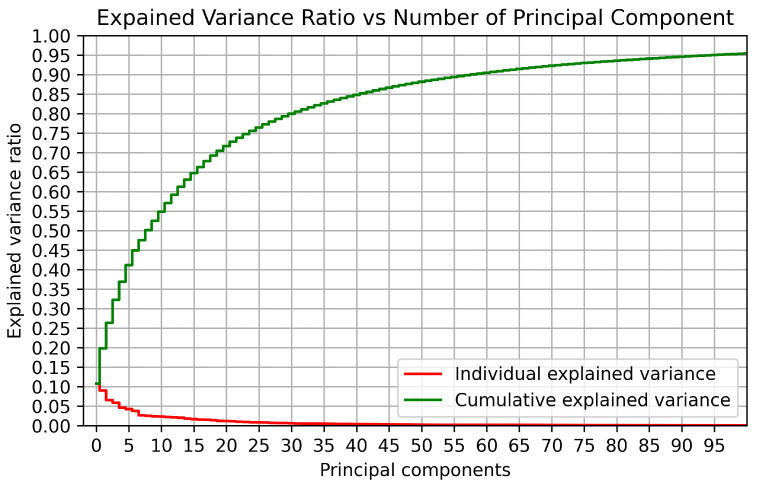
Explained variance ratio (EVR) vs number of principal components for the UJIIndoorLoc dataset.

**Figure 5 sensors-22-06629-f005:**
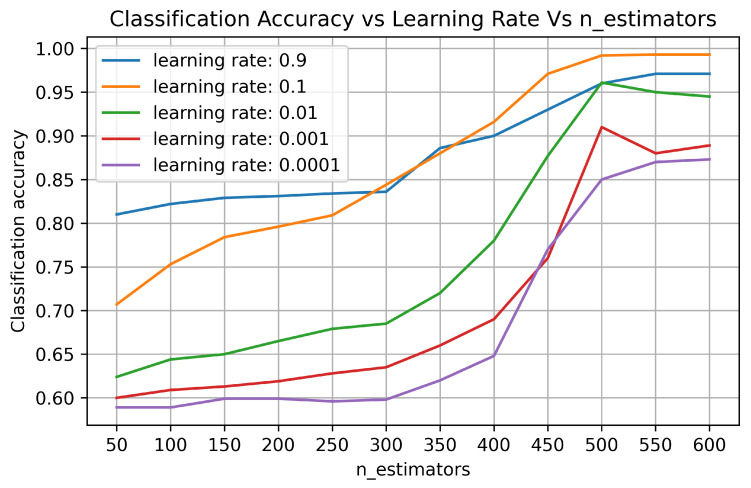
Classification accuracy vs. learning rate vs. n_estimators.

**Table 1 sensors-22-06629-t001:** Summary of recent related works. [Note] P Dataset: public dataset used, H tuning: hyperparameter tuning required, B/F Classification: building and floor classification.

Ref.	P Dataset	Labels	Techniques Used	H Tuning	B/F Classification
[25]	✓	NRL	RNN	Extensive	✓
[26]	✓	NRL	RNN	Extensive	✓
[13]	✓	NRL	SAE, CNN	Extensive	✓
[27]	✓	NRL	DNN	Extensive	✓
[28]	✓	NRL	2D Radio map, CNN	Extensive	✓
[29]	✓	NRL	m-CELL, 2D CNN	Extensive	✓
**XGBLoc**	✓	**RL**	**PCA, XGBoost**	**Less**	✓

**Table 2 sensors-22-06629-t002:** Elements of fingerprints in UJIIndoorLoc.

Elements of Fingerprints (fi)	Description
*R*	$\left(r_{1},r_{2},r_{3},r_{4},\ldots ,r_{i},\ldots ,r_{520}\right)$, and $r_{i}$ is RSSI value of *i*th access point.
long	Longitudinal values of location in meters.
lat	Latitudinal values of location in meters.
fl	Floor ID.
bi	Building ID.

**Table 3 sensors-22-06629-t003:** Symbols used for defining objective function of proposed ML model.

Symbols	Description
*D*	Fingerprint dataset.
*N*	Total number of samples in dataset.
*M*	Total number of dimmensions/WAPs.
$R_{i}$	RSSI vector at *i*th RP.
$l_{i}$	Position of *i*th RP.
$\left[x_{i},y_{i}\right]$	Coordinates of the $l_{i}$.
*K*	Total number of trees.
$\hat{l_{i}}$	Predicted position of *i*th RP.
$\left[\hat{x_{i}},\hat{y_{i}}\right]$	Predicted coordinates of the $\hat{l_{i}}$.
L(·,·)	Loss function.
ω(·)	Regularization term of the *k*th tree.

**Table 4 sensors-22-06629-t004:** XGBLoc hyperparameters.

Hyperparameter	Value	Description
learning_rate or eta	0.3	Weighting factor for learning in gradient boosting.
gamma	1	Minimum loss reduction needed to render partition on a tree leaf node.
max_depth	6	Maximum depth of tree.
colsample_bytree	1	Subsample ratio of columns when constructing each tree.
lambda	1	L2 regularization term on weights.
loss function	multi:softprob reg:squarederror	Multiclass classification problem. Regression with squared loss.
n_estimators	100	Number of trees to be generated.
scale_pos_weight	1	Control the balance of positive and negative weights.
booster	gbtree	Use tree based model.
tree_method	gpu_hist	GPU implementation of faster histogram optimized approximate greedy algorithm.
Subsample	1	Subsample ratio of the training samples.

**Table 5 sensors-22-06629-t005:** Effect of explained variance ratio (EVR) on localization performance with different input and output specification.

Explained Variance Ratio (EVR)	0.7	0.8	0.9
Classification Accuracy	98%	98.6%	99.2%
2-D Mean position error (long,lat)	5.2 m	4.93 m	5.01 m

**Table 6 sensors-22-06629-t006:** Effect of hyperparameter tuning on localization performance. [Note] lr: learning_rate, md: max_depth, cb: colsample_bytree, lf: loss function, ne: n_estimators.

Task	Hyperparameter								Output
	lr	gamma	md	cb	lambda	lf	ne	Subsample	
**Classification**	0.1	1	6	0.9	0.8	multi:softprob	500	0.8	**99.2%**
**Regression**	0.1	0	10	0.8	0.9	reg:squarederror	1000	0.8	**4.93 m**

**Table 7 sensors-22-06629-t007:** Classification results of existing and proposed schemes on the UJIIndoorLoc dataset.

WiFi Fingerprint-Based Schemes	Classification Accuracy	
	Building	Floor
MOSAIC	98.5%	93.83%
1-KNN	100%	89.95%
13-KNN	100%	95.17%
DNN	100%	91.97%
2D-DNN	100%	95.64%
Scalable DNN	99.5%	91.26%
CNNLoc	100%	96.03%
**XGBLoc**	**100%**	**99.20%**

**Table 8 sensors-22-06629-t008:** Regression results of existing and proposed schemes on the UJIIndoorLoc dataset.

WiFi Fingerprint-Based Schemes	2-D Average Positioning Error (m)
KNN	7.9
WKNN	6.2
HybLoc	6.46
RF	10.2
CNNLoc	11.78
CCpos	12.4
**XGBLoc**	**4.93**

**Table 9 sensors-22-06629-t009:** Classification results of existing and proposed schemes on the Tampere dataset.

WiFi Fingerprint-Based Schemes	Classification Results
Weighted Centroid	83.18%
Log-Gaussian Probability	85.30%
RSS Clustering	90.79%
UJI KNN	92.97%
RTLS@UM	90.03%
Rank-based	86.48%
Coverage Area-based	86.56%
CNNLoc	94.12%
**XGBLoc**	**97.03%**

**Table 10 sensors-22-06629-t010:** Regression results of existing and proposed schemes on the Alcala dataset.

WiFi Fingerprint-Based Schemes	2-D Average Positioning Error (m)
KNN	2.62
WKNN	2.27
SVM	6.71
RF	2.53
CNNLoc	4.62
CCpos	1.05
**XGBLoc**	**1.5**

## Data Availability

No new data were created or analyzed in this study. Data sharing is not applicable to this article.

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
