# Peer review of "XGBLoc: XGBoost-Based Indoor Localization in Multi-Building Multi-Floor Environments"

_sensors, 2022, doi:10.3390/s22176629_

Round 1

Reviewer 1 Report

1. What are the limitations of the proposed model ?

2. Better to tabulate the properties of the datasets used in this study in the appendix.

3. Please provide authors comments on reasons for improved efficiency of authors model compare to other models of 

95%+ accuracy in table 5 ?

4. What are the challenges in implementing RL and XGBLoc from any general building or set of buildings or future 

buildings w.r.t. advancement of tech ?  What is the direction of future study ?

5. The list of schemes in tables 5 - 8 are different. Is there a reason for  showing different schemes in each tables.

6. Authors should show try to show novelty of the research in layout of the article presentation.

Author Response

C1. What are the limitations of the proposed model?

Ans: We thank reviewer for the comment. The proposed indoor localization model focuses on the classification/regression of mobile users in multi-building multi-floor environments (MBMF). The UJIIndoorLoc dataset in [18] we have used is just measured from a university with 3 buildings, each having 4 or 5 floors. That is, it is a relatively small scale MBMF but not a large or medium-scale MBMF that we often meet in a big city like Seoul or New York. As a result, evaluation of proposed scheme over the large or medium-scale MBMF is limited since the corresponding such public (or private) dataset is not yet available.

C2. Better to tabulate the properties of the datasets used in this study in the appendix.

Ans: We thank reviewer for the suggestion. In the manuscript, we have used three datasets each representing different type of indoor environments, i.e., UJIIndoorLoc for MBMF [18], Tampere for single building multi-floor (SBMF) [19], and Alcala for single building single floor (SBSF) [20]. Especially, we have summarized the main elements of our main dataset UJIIndoorLoc in Table 2 and stated its other properties in detail in the corresponding text of our new version. In our new manuscript, we have also summarized some properties of the other two datasets (whose details are referred to their references [19, 20]) in Section 4.2.

C3. Please provide authors comments on reasons for improved efficiency of authors model compare to other models of 95%+ accuracy in table 5?

Ans: We thank reviewer for the comment. Rather than non-relational labeling (NRL) that existing models have used, our proposed XGBoost-based model (termed as XGBLoc) employs relational labeling (RL), where a building ID is properly associated with its corresponding floor IDs. XGBoost that is easily dealt with such tabular data learning would be a good candidate for classifying such RL labels. Therefore, the proposed ML model XGBLoc distinguishes building & floor IDs in such complex and hierarchical indoor localization (IL) environments well. Hence, our proposed model has more improved classification performance over existing benchmark models. Another reason is that XGBLoc with PCA can deal with missing and redundant data well without extensive hyperparameter tuning. Therefore, simulation results in Table 5 (now Table 7) show that XGBLoc is superior to existing bench-mark models in terms of classification performance as well as system design efficiency (or system complexity).   

C4. What are the challenges in implementing RL and XGBLoc from any general building or set of buildings or future buildings w.r.t. advancement of tech ?  What is the direction of future study?

Ans: We thank reviewer for the comment. The proposed XGBoost-based scheme, termed as XGBLoc, could easily be implemented in any general building or set of buildings since XGBoost is a scalable classifier. In the manuscript, we have tested the proposed scheme on a multi-building multi-floor environment using UJIIndoorLoc dataset. We have also used Tampere (single building multi-floor) and Alcala (single building single floor) datasets to test the scalability and robustness of the proposed scheme. However, UJIIndoorLoc is just a small-scale MBMF data that is collected from a university with 3 buildings, each having 4 or 5 floors. As a result, in the literature, due to the lack of (public/private) datasets collected from large or medium-scale MBMF, the performance effect of increasing buildings and their corresponding floors on XGBLoc with RL is not fully evaluated. Hence, in near future (as addressed in Section 5), we will expand our study further by collecting data from multiple buildings covering large area and testing the behaviour of XGBLoc on it. Additionally, our work will be further expanded for achieving low-latency user localization, while preserving the user privacy and security, which could help develop the location-based access control and resource management system for smart factories.

C5. The list of schemes in tables 5 - 8 are different. Is there a reason for showing different schemes in each tables.

Ans: We thank reviewer for the comment. In the paper, we have proposed XGBLoc for multi-building multi-floor IL environments (MBMF; using the UJIIndoorLoc dataset), whose classification performance has been shown and compared with corresponding benchmark schemes in Table 5 (now Table 7). And in Table 6 (now Table 8), regression performance of our proposed scheme over MBMF has been shown which is also compared with its corresponding benchmark schemes. Moreover, to test the proposed scheme's adaptability, robustness, and scalability further, we have also used two additional public datasets each collected from different indoor environments, i.e., Tampere from a single building multi-floor (SBMF) and Alcala from a single building single floor (SBSF). In Table 7 (now Table 9), we have shown the classification performance of the proposed scheme over SBMF and compared it with corresponding benchmark schemes. Similarly, in Table 8 (now Table 10), we have shown the regression performance of the proposed scheme over SBSF and compared it with existing schemes. That is, due to use of each dataset representing different (unique) indoor environments, their corresponding benchmark schemes in tables are different.

C6. Authors should show try to show novelty of the research in layout of the article presentation.

Ans: We thank reviewer for the suggestion. First novelty of our presented model XGBLoc is to use relational labels, rather than non-relational labels that existing schemes have used, for the representation of the complex & hierarchical indoor environments. Hence, our system model that is trained using such translated tabular data has an improved classification performance for MBMF. Another novelty of our proposed model is to employ a state-of-art gradient boosting classifier XGBoost for the indoor localization over MBMF. XGBoost is better fitted to such tabular data rather than other ML/DL classifiers that existing models have used. Simulation results prove that XGBLoc is superior to existing ML/DL algorithms (see Table 7 to 10). Additionally, our proposed localization algorithm that includes a preprocessing module PCA deals with over- and under-fitting issues well. As a result, hyperparameter tuning process of the proposed algorithm is relatively simple and has short computational time, when compared to existing ML/DL algorithms (like RF, KNN, and so forth).

Reviewer 2 Report

In terms of the multiple applications in intelligent buildings, the proposal would initially appear very attractive for applied research; however, the paper presented does not show any basic research to support it. Figure 4 does not support the measurement, performance, and behavior of the proposed solution. The following considerations are recommended to the authors.

It is recommended that the authors change the structure of the paper.

1.- Introduction (The general part of the problem supported by scientific articles). It is recommended to elaborate in Illustrator on a conceptual graphic that shows the problem and the proposed solution. Eliminate figure 1, which is no longer necessary.

 2.- Related Works (Bibliometric analysis of works specifically related to the proposal and supported with scientific articles). It is recommended to make a summary table with the most relevant proposals between 2022 and 2019 and compare it with the present work what is the innovation and added value.

3.- Formulation of the Problem and Methodology (A table of variables used in the mathematical model, the mathematical model that supports the proposal, and the pseudocode of the algorithms used should be included).

4.- Analysis of Results (In this section, it is recommended to increase the number of metrics and considerably reduce the number of tables).

5.- Conclusions (To make a cross-check of information between the objective stated in the document's summary with the results found).

6.- References (Use a range between 2022 and 2019). Use articles from ScienceDirect, MDPI, Taylor & Francis, SAGE, Wiley, Springer, Hindawi, and IEEE Xplore [Transactions, Journals, Magazines].

Authors are encouraged to use Overleaf as a word processor and Mendeley as a bibliographic manager to complete incomplete or erroneous data for each reference. The metrics can be elaborated in Matlab, and the command print -dpdf -r800 figure4 obtain the metrics. 

Author Response

C1: In terms of the multiple applications in intelligent buildings, the proposal would initially appear very attractive for applied research; however, the paper presented does not show any basic research to support it. Figure 4 does not support the measurement, performance, and behavior of the proposed solution. The following considerations are recommended to the authors. It is recommended that the authors change the structure of the paper. 1. Introduction (The general part of the problem supported by scientific articles). It is recommended to elaborate in Illustrator on a conceptual graphic that shows the problem and the proposed solution. Eliminate figure 1, which is no longer necessary.

Ans: We thank reviewer for the comment. As per suggestion given by reviewer, we updated the structure of the paper wherever it is possible (please see below answers of C2 to C6 as well). Especially regarding the Section 1. Introduction, we first gave a general introduction including background & motivation for the main topic of the paper “WiFi RSSI fingerprinting-based IL over multi-building multi-floor indoor environments”. We then introduced our proposed scheme XGBLoc overcoming current issues of existing schemes, summarized its main contributions, and finally gave a brief paper organization. In the updated text of Section 1, following reviewer’s comments, we elaborated the general part of the problems of existing schemes further (including recently-published appropriate references [12],[13])—please see Page #2, Second paragraph. However, we did not eliminate Figure 1—actually it is cited in Section 2 instead Section 1, because it provides a graphical view about a main issue of existing ML/DL schemes, i.e., “use of non-relational labelling (NRL)” (see Figure 1[left]). In Figure 1[right], we also graphically explain our proposed solution for that issue, i.e., “use of relational labelling (RL)”, whose detail concept is referred to in Section 3.2.

Our proposed scheme XGBLoc equips a preprocessing function PCA for dimensionality reduction and denoising of the given RSSI data. Figure 4 shows number of principal components (dimensions) vs EVR (explained variance ratio) which explains important representation for a given data. Figure 4 shows that the dimensionality reduces from 520 to 100 by employing PCA for a given data UJIIndoorLoc. It implies that only 100 dimensions/principal components are enough to get 90% useful information present in that dataset and remaining components are either noise or have less importance. This intuition helps us to reduce the dataset dimensionality, which releases a big burden of training and testing for such large-scale localization task.

C2: Related Works (Bibliometric analysis of works specifically related to the proposal and supported with scientific articles). It is recommended to make a summary table with the most relevant proposals between 2022 and 2019 and compare it with the present work what is the innovation and added value.

Ans: We thank reviewer for the suggestion. Following reviewer’s comment, we updated Section 2. Related Works by including most recent & relevant articles from 2019 to 2022 and comparing it with the proposed scheme XGBLoc. Summary table (see Page #4, Table 1) for those existing schemes is added and compared together with XGBLoc. Please note that the shortcomings of those existing schemes are dealt with in Section 2.1 in detail.  Please also look at the updated part of Section 2 in our new version (see Page #3, Lines 102 to 125 and Page #4, Lines 126 to 133).

C3: Formulation of the Problem and Methodology (A table of variables used in the mathematical model, the mathematical model that supports the proposal, and the pseudocode of the algorithms used should be included).

Ans: We thank reviewer for the suggestion. We updated Section 3 and changed its title from “System Model” to “Proposed Methodology”, which is more aligned with reviewer’s suggestion. We also included a table of symbols in Section 3 (See Page #7, Table 3) that are mainly used for defining the objective function of our proposed XGBoost-based localization model, termed as XGBLoc. Additionally, we also included the pseudocode of XGBloc (see Page #10, Figure 5).

C4: Analysis of Results (In this section, it is recommended to increase the number of metrics and considerably reduce the number of tables).

Ans: We thank reviewer for the suggestion. We updated Section 4. Performance Analysis by including Figure 6 (see Page #11) as per reviewer’s comment “added metrics”, which demonstrates the effect of n_estimators (the number of trees that is generated and summed up) on classification accuracy of the proposed scheme. However, we realized that it is not easy to reduce the number of tables since the performance results of the proposed scheme are shown comprehensively as well as distinctively using each of those tables. For instance, Table 5 and 6 show the effect of EVR and Hyperparameter tuning on the performance of proposed scheme, respectively. And Table 7-10 show the comparison between benchmark schemes and proposed scheme on respective dataset representing different indoor environments: UJIIndoorLoc (multi-building multi-floor), Tampere (single building multi-floor), and Alcala (single building single floor).

C5: Conclusions (To make a cross-check of information between the objective stated in the document's summary with the results found).

Ans: We thank reviewer for the suggestion. We checked the conclusion section (Section 5; please see the updated part Page #14, Lines 420 to 425) according to reviewer’s comments.

C6: References (Use a range between 2022 and 2019). Use articles from ScienceDirect, MDPI, Taylor & Francis, SAGE, Wiley, Springer, Hindawi, and IEEE Xplore [Transactions, Journals, Magazines].

Ans: We thank reviewer for the suggestion. Following reviewer’s comment, we have updated the reference section that adds recently-published appropriate articles (including [25-27], [29]) from high quality transactions, journals, conferences, and magazines.

C7: Authors are encouraged to use Overleaf as a word processor and Mendeley as a bibliographic manager to complete incomplete or erroneous data for each reference. The metrics can be elaborated in Matlab, and the command print -dpdf -r800 figure4 obtain the metrics.

Ans: We thank reviewer for the suggestion. We used Overleaf as a word processor and Mendeley as a bibliographic manger. Please note that as mentioned in Section 4. Performance Analysis, our proposed system model (whose simulation outputs including graphic metrics are automatically generated) is implemented with Python 3.8.5.

Reviewer 3 Report

This paper proposed an approach for indoor localization based on extreme gradient-based booting algorithm (XGBoost). Details of the implementation is given and the comparison to the related work is provided. The proposed approach makes use of the structured synthetics labels to improve the localization accuracy. One question is that the proposed approach is designed for multiple building and multiple floor environment. How to apply the proposed approach in a smaller environment, for example one room or one floor of a building? Some other comments can be found below:

1. Typo errors can be found in the paper. For example, line 66, “accuracy the”->“accuracy in the”;

2. In Table 1, the variable sp and rel are not explained in the paper. Are they used for training and localization as well?

3. In Equation 3, how you can get the min value. Do you mean -100?

4. In Section 4.1, what is the difference between testing and validation.

5. In Table 4, what does regression mean here? Do you mean localization? Why classification and regression are using different parameter settings?

6. The approaches listed in Table 5 and Table 7 are quite different. I suggest to use the same approaches for comparison.

7. The following papers are quite related to this paper. I suggest to add them as reference for discussion:

Recent advances in indoor localization: A survey on theoretical approaches and applications. IEEE Communications Surveys Tutorials.

Fusing similarity-based sequence and dead reckoning for indoor positioning without training. IEEE Sensors Journal.

Author Response

This paper proposed an approach for indoor localization based on extreme gradient-based booting algorithm (XGBoost). Details of the implementation is given and the comparison to the related work is provided. The proposed approach makes use of the structured synthetics labels to improve the localization accuracy. One question is that the proposed approach is designed for multiple building and multiple floor environment. How to apply the proposed approach in a smaller environment, for example one room or one floor of a building? Some other comments can be found below:

Ans: We thank reviewer for the comment. The proposed XGBoost-based ML localization scheme is easily applicable for any general hierarchical indoor environments since XGBoost has a good scalability [14]. As seen in the manuscript (see Section 4), we have first tested the proposed scheme on a multi-building multi-floor environment using the UJIIndoorLoc dataset and then experimented it on two smaller indoor environments, i.e., a single building multi-floor environment using the Tampere dataset and a single building single floor using the Alcala dataset. As queried by reviewer, in the case of one room or one floor of a building (which could belong to the single building single floor case like Alcala), our localization scheme focuses on the prediction of position coordinates of mobile users by using latitude & longitude values of related RPs. That is, it pursues an optimal solution for regression problem rather than classification problem. However, the proposed scheme XGBLoc is supposed to give a solution for both regression and classification problems over any general indoor environments.

C1: Typo errors can be found in the paper. For example, line 66, “accuracy the”->“accuracy in the”;

Ans: We thank reviewer for the suggestion. We have done extensive proofreading to improve the quality of English and remove typos (including the one indicated by reviewer) as much as possible.

C2: In Table 1, the variable sp and rel are not explained in the paper. Are they used for training and localization as well?

Ans: We thank reviewer for the comment. As per suggestion given by reviewer, from Table 1 (new Table 2), we have removed those unused variables (sp & rel).

C3: In Equation 3, how you can get the min value. Do you mean -100?

Ans: We thank reviewer for the comment. The min value indicates the minimum of RSSI values y [dBm] of the offline fingerprint database. For instance, when assuming -98 £ y £ 0, then the min value ymin = -98 [dBm]. So the zero-to-one normalization in EQ 3 is achievable.

C4: In Section 4.1, what is the difference between testing and validation.

Ans: We thank reviewer for the comment. Generally, a given data is split into three sets, i.e., training, validation, and test sets. While the training and validation sets are used for the model's training, the test set is used for the model’s testing. Especially, a validation set is used for fine-tuning of model hyperparameters, where the model's performance is validated with different hyperparameter values. After such hyperparameter tuning, the test set is used for the final performance checking of the presented model, which is also called final model fitting.

C5: In Table 4, what does regression mean here? Do you mean localization? Why classification and regression are using different parameter settings?

Ans: We thank reviewer for the comment. Localization is largely divided into two tasks: classification and regression. Regression is to predict the physical location (that is, coordinates) of the devices in the building while classification is to predict the symbolic location of the devices using building IDs and floor IDs. As seen in Table 4 (now Table 6), since goal of those two tasks is different, the parameter setting of achieving optimal localization results could be different. 

C6: The approaches listed in Table 5 and Table 7 are quite different. I suggest to use the same approaches for comparison.

Ans: We thank reviewer for the comment. In the paper, we have proposed a fingerprinting-based indoor localization scheme XGBLoc and evaluated it over different indoor environments.  First, in Table 5 (now Table 7), we have shown the classification performance of the proposed scheme on the UJIIndoorLoc dataset (multi-building multi-floor) and compared it with corresponding benchmark schemes. Second, in Table 7 (now Table 9), we have shown the classification performance of the proposed scheme on the Tampere dataset (single building multi-floor) and compared it with corresponding benchmark schemes. That is, due to the use of each dataset representing different (unique) indoor environments, their corresponding benchmark schemes could be also different. Therefore, Tables 5 and 7 (now Table 7 and 9) show the different list.

C7: The following papers are quite related to this paper. I suggest to add them as reference for discussion:

Recent advances in indoor localization: A survey on theoretical approaches and applications. IEEE Communications Surveys Tutorials.

Fusing similarity-based sequence and dead reckoning for indoor positioning without training. IEEE Sensors Journal.

Ans: We thank reviewer for the suggestion. We have updated the manuscript by adding those two recommended articles (see References [8] and [10]).

Round 2

Reviewer 2 Report

The quality of figures 4 and 6 are deficient.

It is suggested in Matlab

hold on; grid on; box 'on';

print -dpdf -r800 figure4

It is suggested and recommended to extend at least two additional metrics to figures 4 and 6 as this will demonstrate the testing that has been carried out and the algorithm's performance. A single curve does not show a valid contribution to the subject.

It is suggested to review the incomplete or incorrect references' metadata. It is recommended to use Mendeley as a bibliographic manager to correct before taking to Overleaf.

Author Response

C1. The quality of figures 4 and 6 are deficient. It is suggested in Matlab

hold on; grid on; box 'on';

print -dpdf -r800 figure4

 Ans. We thank the reviewer for the suggestion. We have updated Figure 4 and Figure 6 (now Figure 5) by incorporating the necessary changes to improve their quality (see Page #10 and #12).  

C2. It is suggested and recommended to extend at least two additional metrics to figures 4 and 6 as this will demonstrate the testing that has been carried out and the algorithm's performance. A single curve does not show a valid contribution to the subject.

Ans. We thank the reviewer for the suggestion. For Figure 4 (see Page #10), we have used the standard procedure for displaying Explained Variance Ratio (EVR) vs number of principal components. We have updated (clarified) Figure 4, which shows that by applying PCA to the given data, the top 100 principal components alone can explain more than 90% of the variance of the entire data¾please see the curve of cumulative explained variance. It also shows how much each component contributes to the variance out of the total variance in all the dimensions¾please see the curve of individual explained variance. We have also updated Figure 6 (now Figure 5; see Page #12) with additional metrics, which shows the effect of different learning rates (0.9,0.1,0.01,0.001,0.0001) and n_estimators (50-600) on the classification accuracy of the proposed model. Below paragraph with double quotation marks is the correspondingly updated text regarding Figure 6 (now Figure 5) in our new manuscript.

“Fig. 5 shows the effect of different learning rates and n_estimators (which indicates the number of trees to be generated) on the classification accuracy of the proposed model. The overall trend shows that classification accuracy increases with the increase of n_estimators with different learning rates. From Fig. 5, we can confirm that the classification accuracy of XGBLoc using the UJIIndoorLoc dataset is improved as n_estimators increase and converged to its maximum (99.2%), especially when n_estimators ≥ 500 and the learning rate is kept at 0.1.” (see Page #11, last paragraph)

C3. It is suggested to review the incomplete or incorrect references' metadata. It is recommended to use Mendeley as a bibliographic manager to correct before taking to Overleaf.

Ans. We thank the reviewer for the suggestion. We have fully and carefully checked the Reference section in our new manuscript such that incomplete or incorrect references’ metadata is corrected (including Ref. 1; see Page #15).

Reviewer 3 Report

Figure 5 should follow a standard pseudo code procedure. Please check other papers to improve this part. I also suggest to change Figure 5 to Algorithm 1.

Author Response

C1. Figure 5 should follow a standard pseudo code procedure. Please check other papers to improve this part. I also suggest to change Figure 5 to Algorithm 1.

Ans: We thank the reviewer for the comment. We have updated the manuscript by replacing Figure 5 with Algorithm 1, which follows the standard method for writing the pseudo code as suggested by the reviewer (see Page #10).
